# Recombinant single-cycle influenza virus with exchangeable pseudotypes allows repeated immunization to augment anti-tumour immunity with immune checkpoint inhibitors

**Matheswaran Kandasamy[1]\*[†], Uzi Gileadi[1]\*, Pramila Rijal[1], Tiong Kit Tan[1], Lian N Lee[2], Jili Chen[1], Gennaro Prota[1], Paul Klenerman[2], Alain Townsend[1], Vincenzo Cerundolo[1]**

[1]MRC Human Immunology Unit, Weatherall Institute of Molecular Medicine, University of Oxford, Oxford, United Kingdom; [2]Nuffield Department of Medicine and Translational Gastroenterology Unit, Peter Medawar Building, University of Oxford, Oxford, United Kingdom

**\*For correspondence:**
matheswaran.kandasamy@
kennedy.ox.ac.uk (MK);
uzi.gileadi@imm.ox.ac.uk (UG)

**Present address:** [†]The Kennedy Institute of Rheumatology, University of Oxford, Oxford, United Kingdom

**Competing interest:** The authors declare that no competing interests exist.

**Abstract** Virus-based tumour vaccines offer many advantages compared to other antigen-delivering systems. They generate concerted innate and adaptive immune response, and robust CD8[+] T cell responses. We engineered a non-replicating pseudotyped influenza virus (S-FLU) to deliver the well-known cancer testis antigen, NY-ESO-1 (NY-ESO-1 S-FLU). Intranasal or intramuscular immunization of NY-ESO-1 S-FLU virus in mice elicited a strong NY-ESO-1-specific CD8[+] T cell response in lungs and spleen that resulted in the regression of NY-ESO-1-expressing lung tumour and subcutaneous tumour, respectively. Combined administration with anti-PD-1 antibody, NY-ESO-1 S-FLU virus augmented the tumour protection by reducing the tumour metastasis. We propose that the antigen delivery through S-FLU is highly efficient in inducing antigen-specific CD8[+] T cell response and protection against tumour development in combination with PD-1 blockade.

## Editor's evaluation

The authors found out that virus-based tumour vaccines can induce a robust CTL response capable of limiting tumour progression, which is interesting to many researchers who are looking for ways to enhance CTL response to tumour immunity in combination with checkpoint inhibitors.

## Introduction

The fight against cancer remains unfinished as it continues to be a major threat to human life. Tumour antigen-specific strategies such as the dendritic cell (DC) vaccine (*Palucka and Banchereau, 2012*) have been previously investigated to elicit antigen-specific anti-tumour immunity. Such approaches yielded limited success because of the profound tumour-suppressive environment. Recent successes of treatments with immune checkpoint inhibition are impressive but leave many patients unaffected. Hence, multiple approaches are warranted to reverse tumour immunosuppression and induce anti-tumour T cell responses. The generation of anti-tumour immunity, together with the reversal of tumour immune suppression, might be achieved by triggering innate immune receptors, which have evolved to detect pathogen-associated molecular patterns (PAMPs).

Intrinsically immunogenic pathogens with potential pattern recognition receptor (PRR) agonistic functions can induce potent anti-tumour responses (*Shekarian et al., 2017*), and, with the help of genetic engineering, the immunogenic pathogens can be exploited as vectors to deliver tumour-associated antigens (TAAs). Viruses are naturally immunogenic and represent an attractive vehicle for antigen delivery as several studies have shown that antigens expressed by virus are more immunogenic than soluble antigen administered with adjuvant (*Kantor et al., 1992*; *Kass et al., 1999*).

Influenza A virus (IAV) is an interesting candidate for antigen delivery since IAV infection elicits a strong antigen-specific CTL response. We developed previously a pseudotyped replication-deficient influenza A/Puerto Rico/8/34 (PR8) virus, which replicates only in the cell line expressing HA that provides the HA protein on pseudotyped virus particles for their binding to cells (*Powell et al., 2012*); importantly, the HA expressed by the cell line can be derived from any virus subtype. Here, we generated a recombinant S-FLU virus expressing NY-ESO-1 (New York oesophageal squamous cell carcinoma 1) and evaluated its immunogenicity and therapeutic efficacy in preclinical mouse models. NY-ESO-1 is a well-known cancer testis antigen (CTA). Its expression is normally restricted in germ cells, but it is highly dysregulated in some malignant cells (*Thomas et al., 2018*). Given the immune-privileged nature of germline cells, NY-ESO-1 may be therapeutically targeted without substantial risk of immune-related off-targeted effects. We found that intranasal or intramuscular immunization with NY-ESO-1 S-FLU virus elicits a robust NY-ESO-1-specific CTL response and suppresses the NY-ESO-1-expressing tumour development and spontaneous metastasis. Moreover, with anti-PD1 antibody co-administration, NY-ESO-1 S-FLU virus immunization displays an enhanced tumour reduction.

## Results

### NY-ESO-1 S-FLU virus is able to express exogenous protein and induce a [i]potent antigen-specific CD8[+] T cell response

We generated S-FLU virus to express NY-ESO-1 as described previously (*Powell et al., 2012*; *Figure 1A*, left), and in vitro infection with NY-ESO-1 S-FLU virus shows that the influenza virus NP and NY-ESO-1 proteins were both expressed in infected HEK 293T cells, with a preferential localization within the nucleus and the cytoplasm (*Figure 1—figure supplement 1*), respectively. BALB/c mice infected via nasal route with NY-ESO-1 S-FLU virus displayed a higher level of infection on day 2 in lungs with higher frequency of infected EpCAM[+] lung epithelial cells compared to immune cells (identified as CD45[+]) (*Figure 1—figure supplement 2*).

The ability of intranasal infection with NY-ESO-1 S-FLU virus to induce a T cell primary activation/T cell priming was then investigated in draining lymph nodes (dLN). *Figure 1—figure supplement 3* clearly shows an effective T cell proliferation of 1G4 cells only in the groups of mice infected with the NY-ESO-1 S-FLU, which indicates antigen-specific priming at dLN. Next, we sought to determine if our S-FLU virus was able to induce a detectable immune response stimulating the T cell repertoire of a normal immunocompetent mouse (i.e. without any adoptive transfer). *Figure 1B* shows that intranasal infection with NY-ESO-1 S-FLU virus elicits a robust NY-ESO-1-specific CTL response in HHD and BALB/c mice albeit to a lesser magnitude than the CTL responses against the immunodominant Flu epitopes for M1 and NP proteins, respectively. Intranasal infection with NY-ESO-1 S-FLU virus also induces NY-ESO-1-specific CTL response in spleen (*Figure 1—figure supplement 4*) and displays a specific cytotoxic effect in in vivo killing assay (*Figure 1C*).

### Tumour protection following intranasal infection with NY-ESO-1 S-FLU virus

Using syngeneic 4T1 metastatic breast carcinoma (MBC) tumour model, we tested NY-ESO-1 S-FLU virus anti-tumour effect in the following settings: (1) prophylactic, (2) therapeutic, and (3) spontaneous tumour metastasis models. In prophylactic tumour model, immunization with NY-ESO-1 S-FLU virus significantly reduced the number of nodules in lungs. Interestingly, a single infection with NY-ESO-1 S-FLU was sufficient to give significant protection against tumour challenge (*Figure 2A*). In the therapeutic model, a higher number of tumour nodules developed than in the prophylactic model (~30–40% more) and two sequential infections with NY-ESO-1 S-FLU virus significantly reduced the number of tumour nodules in lungs (*Figure 2B*). In the spontaneous tumour metastasis model, primary tumour displayed spontaneous metastasis to lungs at humane end point and mice that received two

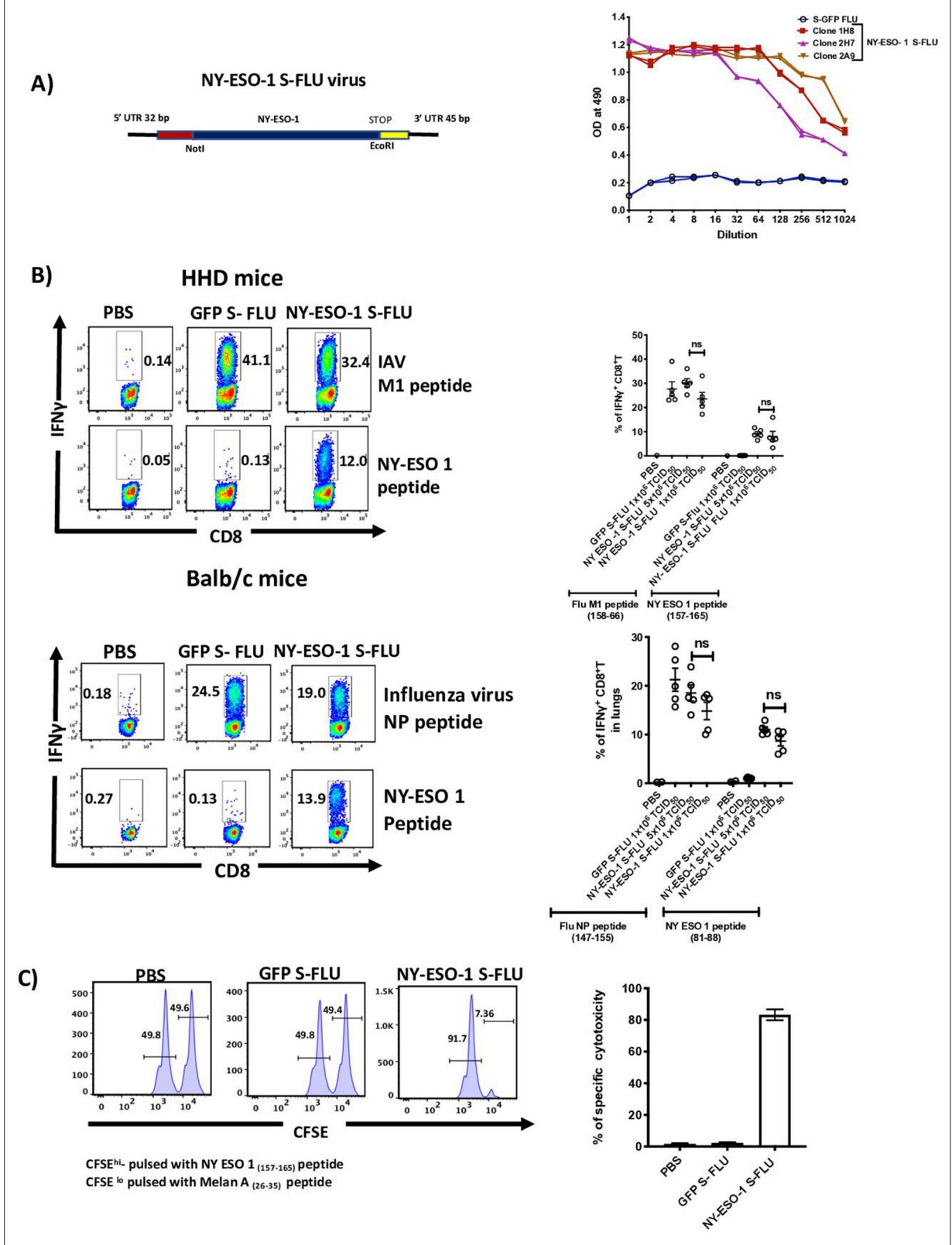

**Figure 1.** Intranasal infection with NY-ESO-1 S-FLU virus elicits a potent CTL response. (**A**) Left: schematic diagram of the design for NY-ESO-1 S-FLU virus. The codon-optimized NY-ESO-1 cDNA sequence between unique NotI site and EcoRI sites in the pPol/S-UL expression cassette surrounded HA packaging sequences. Right: NY-ESO-1 expression in NY-ESO-1 S-FLU-infected MDCK-SIAT1 cells. MDCK-SIA1 cells expressing HA from PR8 were infected with different clones of NY-ESO-1 S-FLU virus or GFP S-FLU virus in twofold serial dilutions and the expression of NY-ESO-1 was analysed after

*Figure 1 continued on next page*

*Figure 1 continued*

48 hr. Three positive S-FLU clones were identified, clone 2A9 was chosen and expanded for further experiments. (**B**) Left: representative flow cytometry dot plots showing the frequency of IFNγ-secreting CD8$^+$ T cells. HHD mice (upper panel) and BALB/c mice (lower panel) were intranasally infected with $1 \times 10^6$ TCID$_{50}$ of GFP S-FLU or $3 \times 10^6$ TCID$_{50}$ and $1 \times 10^6$ TCID$_{50}$ of NY-ESO-1 S-FLU virus on day 0. CTL responses in lungs were analysed on day 10 post infection by ex vivo stimulation with HLA-A2-restricted influenza A virus (IAV) M1 peptide $_{158-66}$(GILGFVFTL) or NY-ESO-1 peptide $_{157-65}$(SLLMWITQC) and H-2K$^d$ binding IAV NP peptide $_{147-55}$(TYQRTRALV) or H-2D$^d$-restricted NY-ESO-1 peptide $_{81-88}$(RGPESRLL). Right panel: bar chart showing the percentage of IFNγ-secreting CD8$^+$ T cells in HHD mice (upper panel) and BALB/c mice (lower panel) lungs. (**C**) In vivo analysis of cytotoxic T cell functions. Left: representative FACS plots for in vivo killing assay. HHD mice were intranasally infected with $1 \times 10^6$ TCID$_{50}$ of GFP- S-FLU or NY-ESO-1 S-FLU virus, and on day 10 post infection, mice received adoptively transferred CFSE-labelled (CFSE$^{hi}$) NY-ESO-1$_{157-65}$ peptide-pulsed splenocytes, and CFSE-labelled (CFSE$^{lo}$) Melan A$_{26-35}$ peptide-pulsed splenocytes, and CFSE-labelled splenocytes were analysed in spleen after 8 hr. Right: bar chart showing the percentage of NY-ESO-1-specific cytotoxicity in infected mice. The values are expressed as mean ± SEM. Data in (**B**) is representative of at least two independent experiments. ns, not significant.

The online version of this article includes the following source data and figure supplement(s) for figure 1:

**Source data 1.** Frequency of antigen specific CD8+T cells in HHD mice.

**Source data 2.** Frequency of antigen specific CTL response in Balb/c mice.

**Source data 3.** In-vivo killing assay.

**Figure supplement 1.** Expression of PR8-NP and NY-ESO-1 proteins in NY-ESO-1 S-FLU virus-infected HEK cells.

**Figure supplement 2.** Intranasally administered NY-ESO-1 S-FLU virus mainly infects lung epithelial cells.

**Figure supplement 3.** In vivo T cell priming at draining lymph node following infection with NY-ESO-1 S-FLU virus.

**Figure supplement 4.** Intranasal infection with NY-ESO-1 S-FLU virus elicits specific CTL response in spleen.

---

sequential infections with NY-ESO-1 S-FLU virus showed fewer colonies (p<0.05) than any of the other treatments (*Figure 2C*).

## Intramuscular administration induces a stronger systemic CTL response than intranasal administration of NY-ESO-1 S-FLU virus

Despite inducing a strong CD8$^+$ T cell response in lungs, intranasal infection did not elicit a high CTL response in spleen (*Figure 1—figure supplement 4*). Furthermore, it failed to protect mice from subcutaneous tumour challenge with NY-ESO-1-expressing 4T1 cells (*Figure 3A*). A strong systemic immune response coordinated across tissues is required for tumour eradication (*Spitzer et al., 2017*). Previously, intramuscular immunization (i.m.) with recombinant adenoviral-vectored vaccine has shown generation of CTL responses in multiple mucosal sites (*Kaufman et al., 2008*).

In order to analyse whether the i.m. route of infection with NY-ESO-1 S-FLU virus would generate a better systemic immune response, NP- or NY-ESO-1-specific CD8$^+$ T cell responses in peripheral blood, lung, and spleen were examined following NY-ESO-1 S-FLU virus infection via intranasal or intramuscular route. In lungs, intranasal infection induced a significantly higher NP (p<0.01) and NY-ESO-1 (p<0.05)-specific CD8$^+$ T cell response compared with intramuscular injection. In the blood and spleen, intramuscular infection induced stronger NP (p<0.01) and NY-ESO-1 (p<0.01)-specific CTL responses than intranasal infection (*Figure 3B*). Similarly, an increased frequency of Flu M1- or NY-ESO-1 specific IFNγ (single producer), or IFNγ and TNFα (double producer) or IFNγ, TNFα, and IL2-producing CD8$^+$ T cells (triple producers) was observed in HHD mice spleen following intramuscular infection (*Figure 3C*), and conversely a higher frequency of flu M1- or NY-ESO-1-specific IFNγ or IFNγ and TNFα-producing CD8$^+$ T cells was detected in HHD mice lungs following intranasal infection (*Figure 3C*). Moreover, intranasal infection and intramuscular infection resulted in the accumulation of higher number of total Flu M1-/NY-ESO-1-specific CD8$^+$ T cells in lungs and spleen, respectively (*Figure 3—figure supplement 1*).

## Intranasal infection induces the expression of tissue retention signals

The reciprocal distribution of antigen-specific CD8$^+$ T cells in lungs and spleen in post intranasal infection indicates that the effector T cells are mostly retained in lungs following intranasal infection. Activated antigen-specific CD8$^+$ T cells have been known to persist following recovery from respiratory virus infection and very late antigen 1 (VLA-1; α1β1 integrin) has been implicated in the retention of T cells (*Hogan et al., 2001*; *Jennrich et al., 2012*; *Ray et al., 2004*).

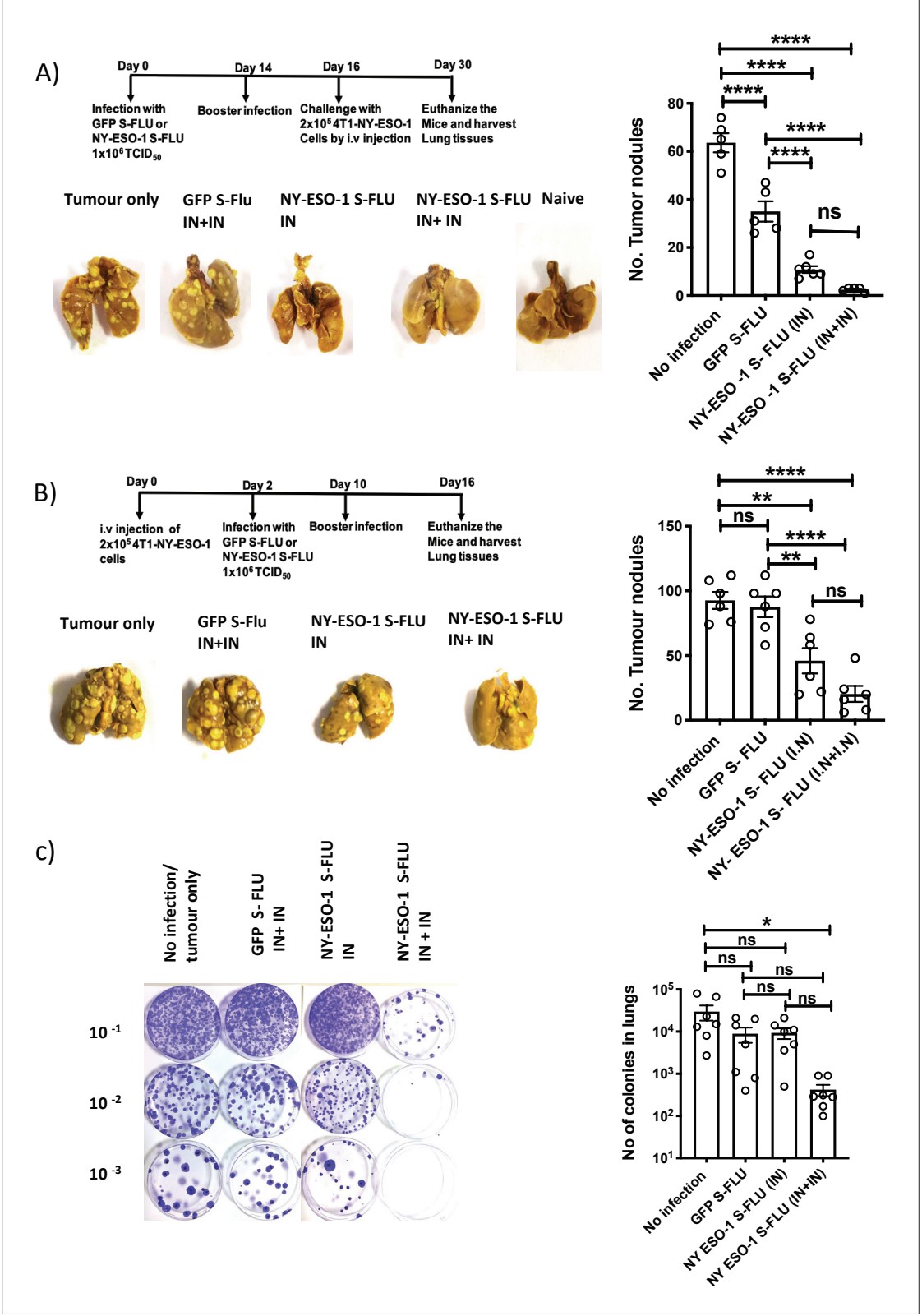

**Figure 2.** Intranasal infection with NY-ESO-1 S-FLU virus protects mice from tumour development in lungs. (**A**) Left: prophylactic tumour model – photographs of the lungs of the mice immunized with NY-ESO-1 S-FLU or GFP S-FLU virus followed by tumour challenge with 4-T1-NY-ESO-1 cells. Right: a bar chart representing the number of tumour nodules enumerated in the lungs. (**B**) Left: therapeutic tumour model – photographs of the lungs of the mice intravenously injected with 4-T1-NY-ESO-1 cells followed by the treatment with NY-ESO-1 S-FLU or GFP S-FLU virus. Right: a bar graph

*Figure 2 continued on next page*

*Figure 2 continued*

representing the number of tumour nodules enumerated in the lungs. Photographs are representative of five mice in two different experiments with similar results. (**C**) Spontaneous tumour metastasis model/clonogenic assay. Left: photographs showing the spontaneous metastasis of 4T1-NY-ESO-1 cells to lungs. 4T1-NY-ESO-1 cells were injected subcutaneously in BALB/c mice, and mice were intranasally infected with GFP S-FLU or NY-ESO-1 S-FLU virus on days 4 and 18 post tumour cell injection. On days 26–28 post tumour cell injection, mice were euthanized and lung cells were plated in 6-thioguonine media. The number of colonies was counted, and the right panel shows the bar chart representing the number of colonies in lungs of the mice treated with GFP S-FLU or NY-ESO-1 S-FLU virus. The results shown in (**A**) and (**B**) are representative of two independent experiments with similar results (n = 5–6 mice/group). Data shown in (**C**) are pooled results of two independent experiments (n = 7 mice/ group). *p<0.05, **p<0.01, ***p<0.001, ****p<0.0001, ns, not significant (one-way ANOVA multiple comparisons). Error bars: mean ± SEM.

The online version of this article includes the following source data for figure 2:

**Source data 1.** Number of tumour nodules.

**Source data 2.** Number of tumour nodules.

**Source data 3.** Number of colonies.

We therefore investigated whether the intranasal or intramuscular infection with NY-ESO-1 S-FLU virus differentially induced the expression of the molecules (VLA-1 [CD49a], CXCR6, CD103, CD44, CD49d, CCR5 and CXCR3) implicated in retention or trafficking of effector CD8+ T cells in lung. The expressions of VLA-1, CXCR6, CD103, and CD44 (p<0.0001) were significantly upregulated on NY-ESO-1-specific CD8+ T cells in the lungs of intranasally infected mice (*Figure 3—figure supplement 2A*) Furthermore, the frequencies of tissue-resident memory CD8+ T cells (TRM) (p<0.0001) were also significantly higher in intranasally infected mice (*Figure 3—figure supplement 2B*).

## Swapping the coat HA in NY-ESO-1 S-Flu virus overcomes the inhibitory effect of pre-existing neutralizing antibodies

NY-ESO-1 S- FLU virus can be pseudotyped with any HA in the envelope and coating with relatively novel HA, against which antibodies are scarcely presented in the population, could nullify the pre-existing antibody mediated effect. To validate that hypothesis, mice were first infected via the intranasal route with X31 (H3N2) virus to elicit the antibody (against H3N2) production, and on day 24 post infection, mice were reinfected with NY-ESO-1 S-FLU virus with matched HA/NA (NY-ESO-1 S-FLU [X31]) or mismatched HA/NA (NY-ESO-1 S-FLU [H1 PR8]). NY-ESO-1-specific CD8+ T cell responses in lungs or spleen were analysed on day 7. NY-ESO-1- and NP-specific CTL responses were reduced in lungs following the infection with NY-ESO-1 S-FLU virus with matched HA/NA (*Figure 4A and B*). Similarly in spleen, infection with NY-ESO-1 S-FLU virus with different HA coating (NY-ESO-1 S-FLU [PR8]) elicited stronger NP-specific CTL response (p<0.0001). However, NY-ESO-1-specific CTL response in spleen was reduced in both matched or mismatched HA/NA NY-ESO-1 S- FLU virus infections (*Figure 4—figure supplement 1*).

## Intramuscular injection of NY-ESO-1 S-FLU virus induces a higher recruitment of NY-ESO-1-specific CD8+ T cells at tumour site and reduces tumour burden

To evaluate the effectiveness of intramuscular infection in tumour-bearing mice, mice with 4T1-NY-ESO-1-established subcutaneous tumour were intramuscularly or intranasally administered with NY-ESO-1 S-FLU virus. Mice treated with intramuscular injection showed a greater reduction in tumour size (p<0.05 on days 18, 20, and 24) and (p<0.01 on days 26 and 28), which was concomitant with significantly higher infiltration of NY-ESO-1-specific CD8+ T cells in TIL (p<0.01) (*Figure 5A*), whereas mice infected via the intranasal route did not show any difference in tumour size (*Figure 5A*). We also investigated the distribution of NY-ESO-1-specific CD8+ T cells in lungs and spleen, and strikingly a higher number of NY-ESO-1-specific CD8+ T cells still accumulated in lungs in intranasally infected mice. However, there was no difference in the distribution of NY-ESO-1-specific CD8+ T cells in spleen of intranasally or intramuscularly infected mice (*Figure 5B*). In clonogenic assay, the number of colonies in lungs following intranasal or intramuscular infection was not significantly different from each other but lower than the mice with no infection (p<0.001) (*Figure 5C*). However, reduced metastases to spleen were only observed in intramuscularly infected mice as depicted in *Figure 5C*.

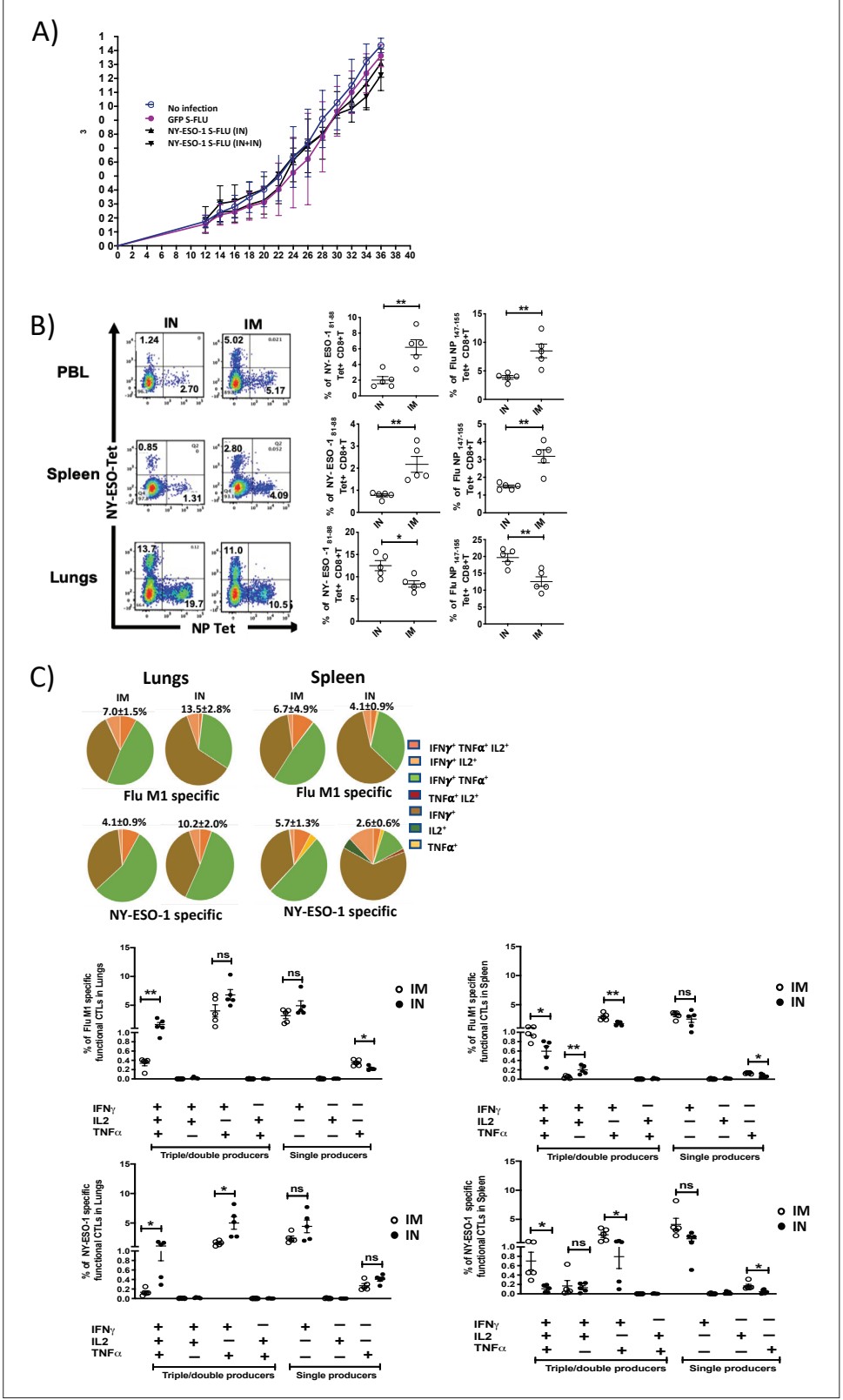

**Figure 3.** Intranasal infection with NY-ESO-1 S-FLU virus fails to protect the mice from subcutaneous tumour challenge and elicits only a modest systemic T cell response. (**A**) Tumour growth of 4T1-NY-ESO-1-bearing mice. BALB/c mice were subcutaneously injected at right flank with $2 \times 10^5$ cultured 4T1-NY-ESO-1 cells on day 0. On days 4 and 18, mice were intranasally infected with GFP S-FLU or NY-ESO-1 S-FLU virus. Tumour volumes were

*Figure 3 continued on next page*

*Figure 3 continued*

monitored every other day. (**B**) NP- or NY-ESO-1-specific CTL responses in blood, lungs, and spleen following intranasal or intramuscular infection. BALB/c mice were intranasally or intramuscularly infected with NY-ESO-1 S-FLU virus, and PR8-NP- or NY-ESO-1-specific CTL responses were analysed by tetramer staining. Left: representative FACS plots showing the percentage of NP- or NY-ESO-1-specific CD8[+] T cells in peripheral blood leukocytes (PBL), spleen, and lungs. Right: the bar graphs display the frequency of NP- or NY-ESO-1-specific CD8[+] T cells. (**C**) Top panel shows the pie charts representing the frequencies of Flu M1-specific or NY-ESO-1-specific polyfunctional CD8[+] T cells (triple cytokines or double cytokine producers) in lungs or spleen. HHD mice were intranasally or intramuscularly infected with NY-ESO-1 S-FLU virus and Flu M1-specific or NY-ESO-1-specific polyfunctional CD8[+] T cells in lungs (left) and spleen (right) were analysed by ex vivo peptide stimulation on day 10 post infection. Middle panel (Flu-M1 specific) and lower panel (NY-ESO-1 specific) show the bar graphs displaying the frequencies of polyfunctional CD8[+] T cells in lungs (left) and spleen (right). The results shown are representative of two independent experiments with similar results (n = 4–5 mice/group). *p<0.05, **p<0.01, ns, not significant (two tailed Student's *t*-test). Error bars: mean ± SEM.

The online version of this article includes the following source data and figure supplement(s) for figure 3:

**Source data 1.** Frequency of polyfunctional NY-ESO-1 specific CD8+T cells in spleen.

**Source data 2.** Frequency of polyfunctional NY-ESO-1 specific CD8+T cells in lungs.

**Source data 3.** Frequency of polyfunctional FLU-M1 specific CD8+T cells in spleen.

**Source data 4.** Frequency of polyfunctional FLU M1 specific CD8+T cells in lungs.

**Source data 5.** Frequency of FLU NP specific CD8+T cell response in Balb/c mice.

**Source data 6.** Frequency of NY-ESO-1 specific CD8+T cell response in Balb/c mice.

**Source data 7.** Frequncy of FLU-NP specific CD8+T cell reponse in spleen.

**Source data 8.** Frequency of NY-ESO-1 specific CD8+T cell response in spleen.

**Source data 9.** Frequency of NP specific CD8+T cell responnse in PBL.

**Source data 10.** Frequency of NY-ESO-1 specific CD8+T cell response in PBL.

**Figure supplement 1.** Absolute number of NY-ESO-1-specific CD8[+] T cells in lungs and spleen following intranasal or intramuscular infection.

**Figure supplement 2.** Intranasal infection induces tissue retention signals and generates a higher frequency of tissue-resident memory CD8[+] T cells.

**Figure supplement 3.** Gating strategy for flow cytometric analysis of lung tissue-resident memory T cells.

## Intramuscular injection with NY-ESO-1 S-FLU virus induces a stronger anti-tumour response than other contemporary virus-based vaccines

Several viruses have been exploited as vehicles for delivering cancer antigens, and we analysed the CTL responses induced by the widely employed or clinically tested viral vectors, based on adenovirus (*Zhang et al., 2003*; *Neukirch et al., 2019*) and Fowl Pox virus (*Townsend et al., 2017*; *Chen et al., 2015*; *Jäger et al., 2006*) that express full-length NY-ESO-1 protein. Infections with recombinant NY-ESO-1 S-FLU and Hu-Ad5-NY-ESO-1 viruses display comparable CTL responses in lungs and spleen, whereas Fowl Pox infection did not elicit a stronger NY-ESO-1-specific CTL response (*Figure 5—figure supplement 1*). Mice treated with NY-ESO-1 S-FLU virus infection displayed a reduced tumour growth (716.9 ± 125.8 mm$^3$) compared to the mice treated with Hu Ad5-NY-ESO-1 (972.0 ± 90.6 mm$^3$) or Fowl Pox-NY-ESO-1 (1137.4 ± 223.6 mm$^3$) viruses (*Figure 6A*). Moreover, mice that received NY-ESO-1 S-FLU virus demonstrated a lower number of 4T1-NY-ESO-1 clones derived from metastatic niches in lungs or spleen in clonogenic assay compared to the mice treated with Hu Ad5 NY-ESO-1 or Fowl Pox virus infection (*Figure 6B*).

## Intramuscular injection of NY-ESO-1 S-FLU virus induces a long-term protection against tumour challenge in HHD mice

To evaluate whether the infection with NY-ESO-1 S-FLU virus could induce a long-term protection against tumour, we have performed a tumour challenge experiment in HHD mice with NY-ESO-1-expressing syngeneic tumour cells (1F4). HHD mice showed a robust NY-ESO-1-specific CTL response following infection with NY-ESO-1 S-FLU virus (*Figure 1B*), and for the tumour challenge experiment, mice were first infected with GFP-S-FLU or NY-ESO-1 S-FLU virus and challenged with subcutaneous

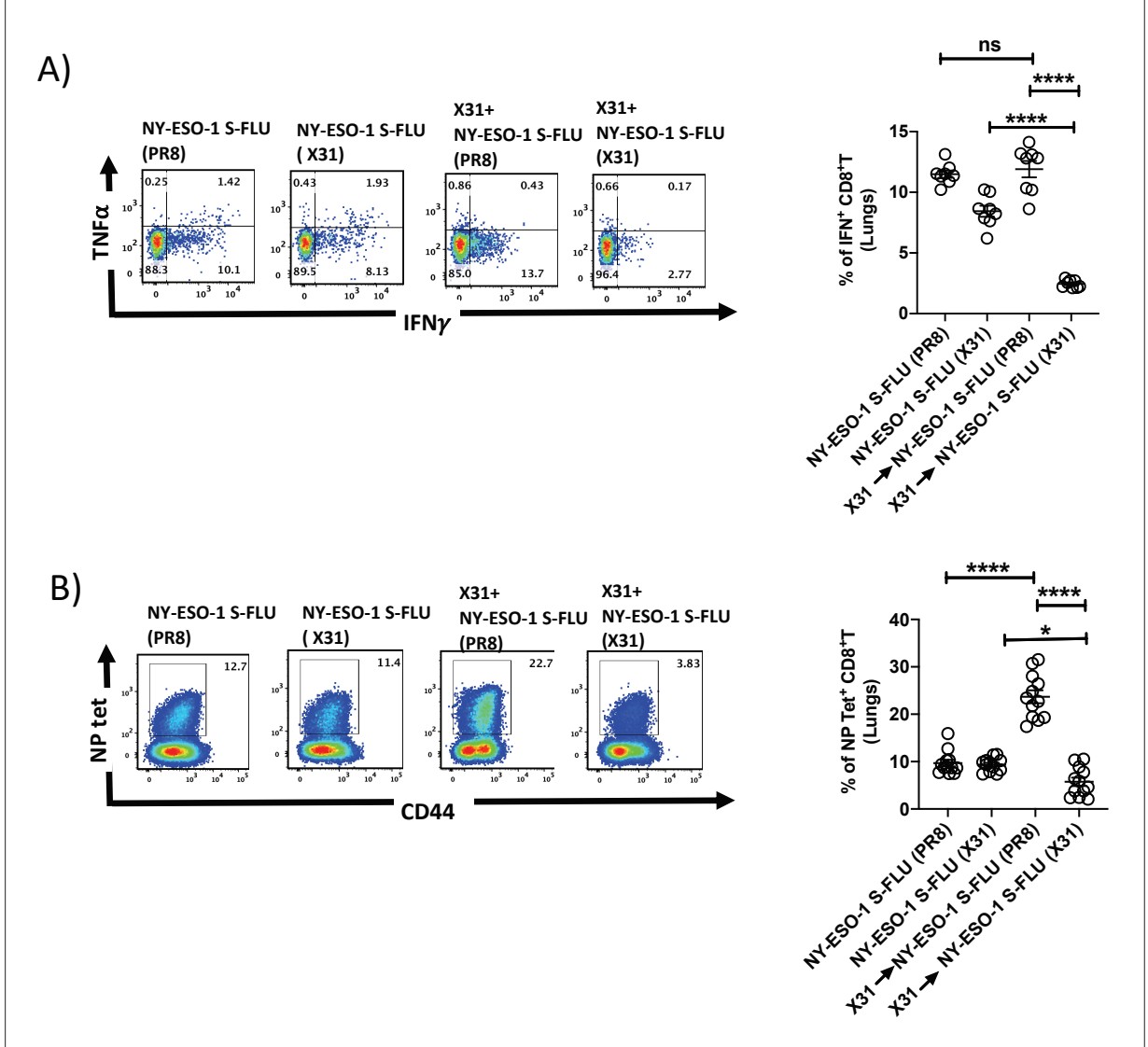

**Figure 4.** Infection with HA-switched NY-ESO-1 S-FLU virus overcomes the inhibitory effect of pre-existing neutralizing antibody. (**A**, left) Representative FACS plot showing IFNγ- and TNFα-secreting and (**B**, left) NP tetramer-positive CD8+ T cells in lungs. Bar graph shows the quantification of IFNγ-secreting CD8+ T cells (**A**, right) and NP tetramer-positive CD8+ T cells (**B**, right) in lungs. BALB/c mice were intranasally infected with X31 virus and reinfected with NY-ESO-1 S-FLU (X31) or NY-ESO-1 S-FLU (PR8) on day 24 post infection, and NY-ESO-1- and NP-specific CD8+ T cell responses in lungs were analysed on day 7 post secondary infection by ex vivo stimulation with NY-ESO-1 CTL peptide $_{81-88}$(RGPESRLL) and NP tetramer staining, respectively. Data shown in (**A**) and (**B**) are pooled results of two (n = 8 mice/group) and three independent experiments (n = 12 mice/group), respectively. *p<0.05, ****p<0.0001, ns, not significant (one-way ANOVA multiple comparisons). Error bars: mean ± SEM.

The online version of this article includes the following source data and figure supplement(s) for figure 4:

**Source data 1.** Frequency of NY-ESO-1 specific IFNg+ CD8+T cells in lungs.

**Source data 2.** Frequency of NP specific IFNg+ CD8+T cells.

**Figure supplement 1.** Infection with HA-switched NY-ESO-1 S-FLU virus elicited a stronger NP-specific CTL responses in spleen.

tumour on day 30 post infection (***Figure 6—figure supplement 1***). Mice treated with intramuscular injection of NY-ESO1 S-FLU showed a greater protection against tumour challenge by displaying reduced tumour size compared to intranasal infection with NY-ESO-1 S-FLU virus or intramuscular injection with GFP-S-FLU virus or the mice with no infection (***Figure 6—figure supplement 1***).

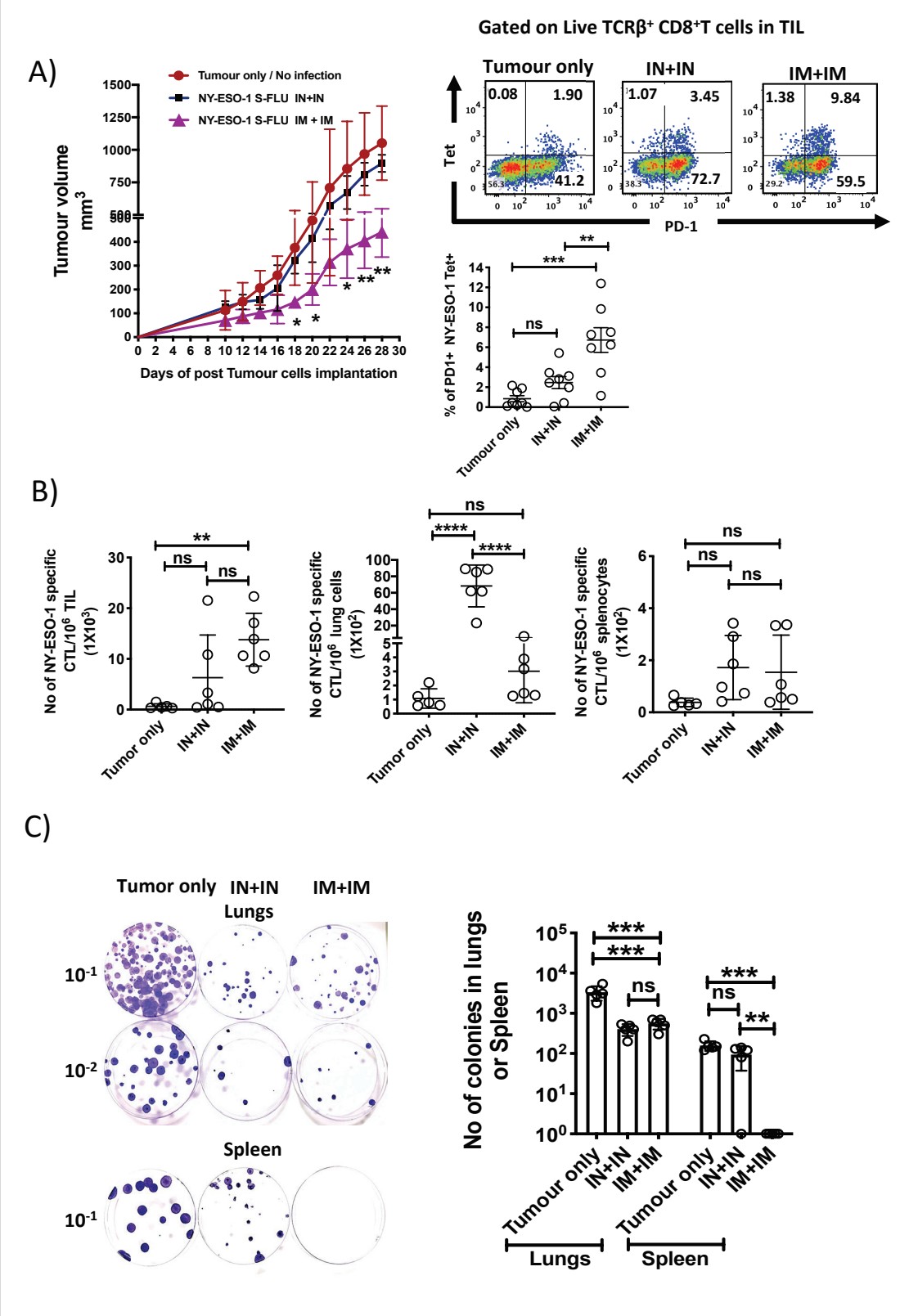

**Figure 5.** Intramuscular injection of NY-ESO-1 S-FLU virus induces a higher infiltration of NY-ESO-1-specific CD8⁺ T cells at tumour site and reduces tumour burden. (**A**) Intramuscular infection with NY-ESO-1 S-FLU virus reduces tumour burden. BALB/c mice were subcutaneously injected with 4T1-NY-ESO-1 cells, and day 4 post injection, mice were intranasally or intramuscularly infected with NY-ESO-1 S-FLU virus followed by booster infection on day 18. Tumour growth was monitored over time. Tumour volume measured 28 days post inoculation in uninfected vs. intranasal infection vs. intramuscular

*Figure 5 continued on next page*

*Figure 5 continued*

infection is shown (left). Right: the representative FACS plots of PD-1$^+$ NY-ESO-1 tetramer$^+$ CD8$^+$ T cells infiltrating the tumour. Bar graph shows the frequencies of PD-1$^+$ Tet$^+$ CD8$^+$ T cells in tumour-infiltrating leukocytes (TIL) in the mice received no infection or intranasal or intramuscular infection with NY-ESO-1 S-FLU virus. (**B**) Bar graphs show the number of NY-ESO-1 tetramer+ CD8$^+$ T cells in TIL or lungs or spleen in tumour-bearing mice received no infection or intranasal or intramuscular infection. (**C**) Clonogenic assay: photographs (left) showing the colonies which represent the spontaneous metastasis of 4T1-NY-ESO-1 cells to lungs. On day 28 post tumour cells inoculation, mice were euthanized followed by digestion of lungs and spleen, and the cells were plated in 6-thioguonine media. The number of colonies was counted after 14 days of incubation. The bar chart (right) represents the number of colonies in lungs and spleen. The results shown are representative of two independent experiments with similar results (n = 4–6 mice/group). Data shown in bar graph in (**A**) are pooled results of two independent experiments (n = 8 mice/ group). *p<0.05, **p<0.01, ***p<0.001, ****p<0.0001, ns, not significant (one-way ANOVA multiple comparisons). Error bars: mean ± SEM.

The online version of this article includes the following source data and figure supplement(s) for figure 5:

**Source data 1.** Tumour volumes (mm3) measured.

**Source data 2.** Frequency of PD1+ NY-ESO-1 specific CD8+T cells in tumour.

**Source data 3.** Frequency of NY-ESO-1 specific CD8+T cells in spleen in tumour bearing mice.

**Source data 4.** Frequency of NY-ESO-1 specific CD8+T cells in TLL in tumour bearing mice.

**Source data 5.** Frequency of NY-ESO-1 specific CD8+T cells in lungs in tumour bearing mice.

**Source data 6.** Number of colonies in lungs or spleen.

**Figure supplement 1.** T cell responses elicited by intramuscular injection of different virus vaccines.

## Blockade of PD1 augments the anti-tumour effect of NY-ESO-1 S-FLU infection

We observed that cytotoxic T cells infiltrated to the tumour displayed a higher expression level of PD1 (*Figure 5A*). To test the possible immune checkpoint role of PD-1 in this setting and improve the effectiveness of NY-ESO-1 S-FLU virus-mediated anti-tumour responses, a combined treatment of NY-ESO-1 S-FLU virus and anti-PD1 antibody was tested. In general, 4T1-NY-ESO-1 tumours positively responded to both anti-PD1 monotherapy and combined therapy with NY-ESO-1 S-FLU virus infection by displaying a slower tumour progression compared to isotype antibody treatment (*Figure 7B*, left). The most pronounced tumour regression was observed in combined treatment with anti-PD1 antibody and intramuscular injection with NY-ESO-1 S-FLU virus (*Figure 7B*, left). The anti-PD1 antibody administration did not increase the infiltration of the NY-ESO-1-specific CD8$^+$ T cells into the tumour in NY-ESO-1 S-FLU virus-infected mice (*Figure 7B*, right), but it was associated with increased expression of CD103 on NY-ESO-1-specific CTLs in TIL (*Figure 7B*, right). In clonogenic assay, anti-PD1 antibody monotherapy showed a reduced tumour metastasis to lungs and notably; in combined therapy with NY-ESO-1 S-FLU virus infection, it demonstrated a significantly higher reduction in tumour metastasis (*Figure 7C*) (p<0.05).

## Discussion

Among many types of influenza vaccines, live-attenuated influenza vaccine (LAIV) is one of the strongest inducers of CD8$^+$ T cell response (*Korenkov et al., 2018*), and the S-FLU virus used in this study is similar to LAIV in terms of generating T cell mucosal immunity in mice and ferrets (*Powell et al., 2012*; *Nogales et al., 2016*). Intranasal immunization of NY-ESO-1 S-FLU virus induced a robust specific CTL response in lungs but with a modest systemic CTL response in spleen. The route of vaccine administration influences the intensity of the systemic antigen-specific T cell response. H5N1 whole-inactivated virus (WIV) immunization via intranasal and intramuscular route induced a comparable frequency of multifunctional Th1 CD4$^+$ cells (*Trondsen et al., 2015*), whereas PR8 WIV strain infection induced a higher IFNγ-secreting CD4$^+$ T cells in spleen (*Bhide et al., 2019*) only in intranasal route. We observed a significantly higher NY-ESO-1-specific CD8$^+$ T cells in blood and spleen with intramuscular immunization, which is in line with a previous study (*Budimir et al., 2013*).

An orthotopic 4T1 mouse breast cancer model used in this study resembles triple-negative breast tumours (ER$^-$, PR$^-$, HER2$^-$) in humans (*Luo et al., 2020*), and we have used 4T1 tumour cells that express a lower level of NY-ESO-1, which may be comparable to the physiological level of NY-ESO-1 expression in human cancer. Intranasal infection with NY-ESO-1 S-FLU virus protected the mice from lung metastasis induced by tail vein or subcutaneous injection. Interestingly, S-FLU without NY-ESO-1 showed a

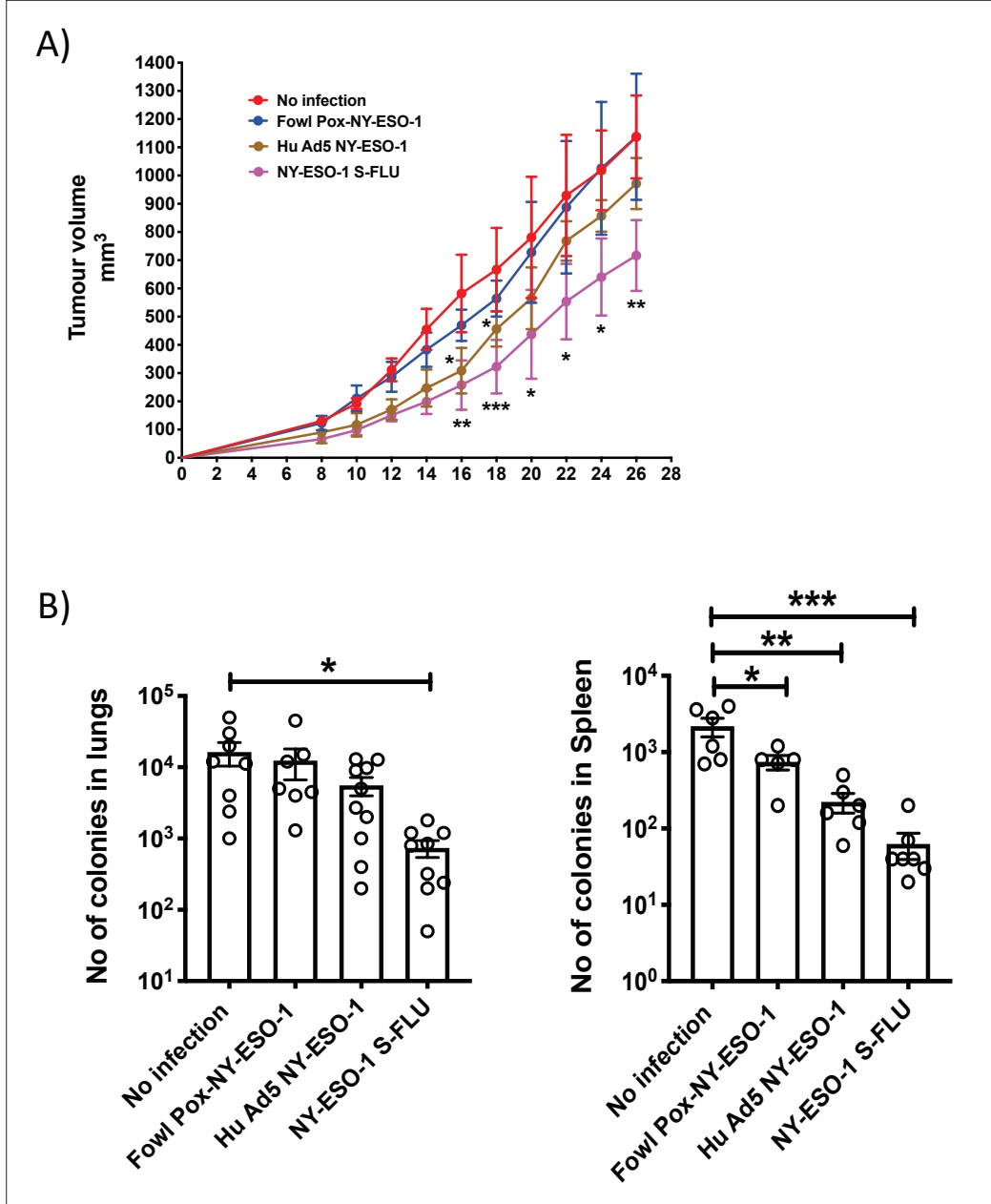

**Figure 6.** Comparison of the anti-tumour effect induced by NY ESO-1 S-FLU virus with contemporary recombinant virus vaccines. (**A**) Intramuscular injection with NY-ESO-1 S-FLU virus demonstrates a better tumour reduction than Hu Ad5 NY-ESO-1 virus or Fowl Pox NY-ESO-1 virus. BALB/c mice were subcutaneously injected with 4T1-NY-ESO-1 cells, and on day 4 post injection, mice were intramuscularly infected with $5 \times 10^7$ PFU NY-ESO-1-expressing Fowl Pox virus or $1 \times 10^9$ PFU human adenovirus 5 or $6 \times 10^6$ TCID$_{50}$ NY-ESO-1 S-FLU virus followed by same dose booster infection on day 18. Tumour growth was monitored over time. Tumour volume measured 28 days post inoculation in uninfected vs. NY-ESO-1 S-FLU vs. Hu Ad5 NY-ESO-1 vs. Fowl Pox NY-ESO-1 virus infection. (**B**) Clonogenic assay: bar graphs showing the number of colonies which represent spontaneous metastasis of 4T1-NY-ESO-1 cells to lungs (left) and spleen (right) in different virus infections described in (**A**). *p<0.05, ** p<0.01, ***p<0.001, ns, not significant (one-way ANOVA multiple comparisons). Error bars: mean ± SEM.

The online version of this article includes the following source data and figure supplement(s) for figure 6:

**Source data 1.** Tumour volumes measured (mm3).

**Source data 2.** Number of colonies in lungs.

**Source data 3.** Number of colonies in spleen.

*Figure 6 continued on next page*

*Figure 6 continued*

**Figure supplement 1.** Memory T cell responses elicited by intramuscular injection of NY-ESO-1 S-FLU protect the mice against tumour challenge.

partial but significant protection against tumour development in lungs (*Figure 2A*), suggesting that non-specific innate immune response mediated through TLR7 by virus single-strand RNA may be utilized for anti-tumour response (*Chi et al., 2017*). Nonetheless, full reduction of spontaneous metastasis was only achieved with NY-ESO-1 S-FLU virus infection, but without any impact on subcutaneous tumour development. Regional tumour protection with intranasal infection may be attributed to the continued presence of antigen-specific CTLs in lungs, which was associated with poor infiltration of CTLs into tumour. Recently, it has been shown that intranasal infection with influenza virus accelerated melanoma growth in skin with increased shunting of anti-tumour CD8$^+$ T cells from the tumour site (skin) to distant site (lungs), resulting in decreased immunity within the tumour (*Newman et al., 2020*). Moreover, increased upregulation of tissue retention molecules VLA-1, CXCR6, CD103, and CD44 on lung CD8$^+$ T cells following intranasal infection (*Figure 6—figure supplement 1A*) probably prevents the T cell migration to distal subcutaneous tumour site, thus permitting the uninhibited tumour growth. In contrast, intramuscular injection with NY-ESO-1 S-FLU virus reduced primary tumour burden with decreased spontaneous tumour metastasis to lungs and spleen (*Figure 5C*). Intriguingly, primary tumour was highly infiltrated with antigen-specific CD8$^+$ T cells but not in lungs or spleens, which suggests that decreased metastasis was likely due to primary tumour regression.

As reported before (*Soboleski et al., 2011*), CTL responses induced with S-FLU virus and recombinant adenovirus infection were comparable (in lungs and spleen); however, further investigation is needed to understand the differential cellular infiltration at tumour site following immunization with S-FLU or adenovirus and whether both viral infections induce a similar frequency and quality (polyfunctionality or tumour cell-killing effect) of tumour-infiltrating antigen-specific CD8+ T cells following infection. With a booster immunization, S-FLU virus induced a more efficient tumour regression than recombinant adenovirus (*Figure 6*). The weaker efficacy in tumour reduction with adenovirus may be due to anti-vector-neutralizing antibodies raised from primary infection (*Sumida et al., 2004*). It has also been highlighted recently that the cellular immune response was not further enhanced following booster immunization with chimpanzee adenovirus-vectored vaccine (ChAdOx1 nCoV-19) for SARS-CoV-2 (*Folegatti et al., 2020*), and options like using different adenoviral vectors for the booster immunization or extending the time more than 10 months between two inoculations have been suggested (*Sayedahmed et al., 2018*). Pre-existing influenza virus immunity may also limit the immunogenicity of influenza-based viral vaccine. Most humans probably have reactive antibodies across different strains by natural infection or flu vaccines, and the extent of broadly reactive antibodies can vary in different individuals depending on the type of virus infection. Limitation of the NY-ESO-1 S-FLU immunization by a pre-existing FLU virus immunity can be overcome, at least partly, by pseudotyping NY-ESO-1 S-FLU with a relatively novel HA with limited exposure in the majority of the population as shown in the proof of principle experiment in *Figure 4*.

Suppression of T cell function is mediated through PD-1 signalling with its ligands PD-L1 and PD-L2 in murine (*Zhang et al., 2009*), patient tumours (*Sato et al., 2005*), and the successful reduction of the tumours has been achieved with checkpoint inhibitors (*Lin et al., 2015*; *Andrews et al., 2019*; *Errico, 2015*; *Tumeh et al., 2014*; *Philippou et al., 2020*). Recently, an enhanced anti-tumour effect has been reported in adenovirus-based vaccine co-administered with immune checkpoint inhibitor (*McAuliffe et al., 2021*). In our study, treatment with anti-PD-1 antibody showed a modest tumour reduction on its own, while the administration in combination with intramuscular delivery of NY-ESO-1 S-FLU virus displayed a synergistic effect with a drastic reduction in tumour size. We administered anti-PD-1 antibody three times only (*Figure 7A*) because of the possibility of hypersensitive reaction in 4T1 tumour-bearing mice (*Mall et al., 2016*), and in future, local delivery of anti-PD-1 antibody may be considered, which would allow more frequent administrations. In a combination therapy, CD103 expression was upregulated on cytotoxic T cells, and it is correlated with tumour reduction as CD103 on CTLs improves TCR antigen sensitivity and enables faster cancer recognition and rapid anti-tumour cytotoxicity (*Qu et al., 2020*; *Abd Hamid et al., 2020*). It would be interesting to investigate in future whether CD103 expression on CTLs is associated with E-cadherin or ICAM-1 expression on tumour

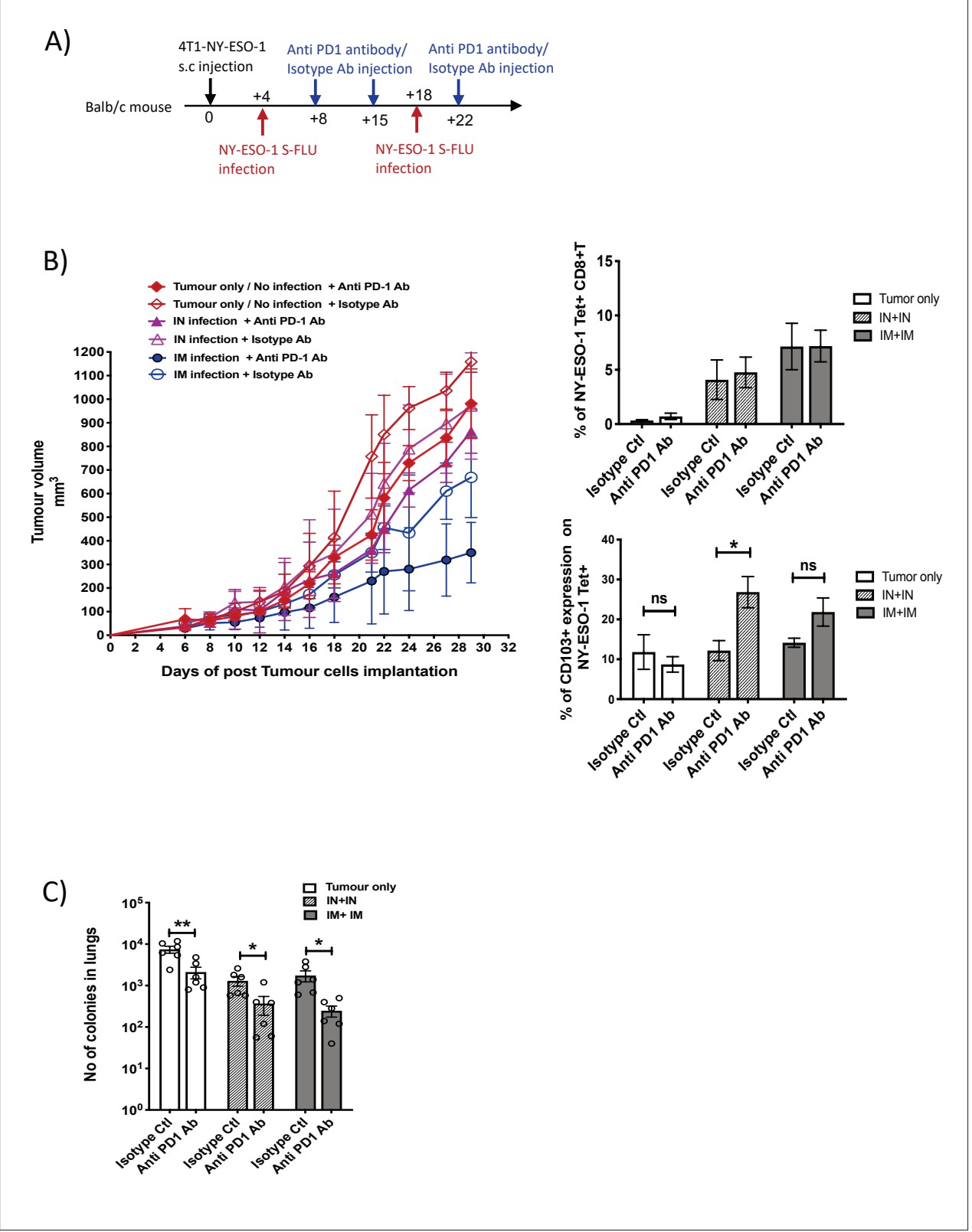

**Figure 7.** Blockade of PD-1 augments the anti-tumour effect of NY ESO-1 S-FLU virus infection. (**A**) Experimental design. (**B**) Tumour growth curves from the experiment described in (**A**) (left) and bar graphs show the frequency of NY-ESO-1 tetramer+ CD8+ T cells in TIL (right upper) and the percentage of CD103 expression on NY-ESO-1 tetramer+ CD8+ T cells in tumour-infiltrating leukocytes (TIL) (right lower). (**C**) Clonogenic assay: bar chart shows the

*Figure 7 continued on next page*

*Figure 7 continued*

number of colonies in lungs. The results shown in **Figure 7** are representative of two independent experiments with similar results (n = 5–6 mice/group). *p<0.05, **p<0.01, ns, not significant (two-tailed Student's *t*-test). Error bars: mean ± SEM.

The online version of this article includes the following source data for figure 7:

**Source data 1.** Tumour volumes measured (in mm3).

**Source data 2.** Frequency of NY-ESO-1 specific CD8+T cells measured by tetramer staining.

**Source data 3.** Number of colonies in lungs.

**Source data 4.** Percentage of CD103 expression on NY-ESO-1 tetramer positive CD8+T cells in tumour.

cells and strong adhesion between the molecules expressed by tumour cells and CD103 on CTLs is required for efficient tumour reduction (*Corgnac et al., 2020*).

In conclusion, this study outlines the efficacy of S-FLU-based tumour vaccine, which displays tumour protection in lungs via intranasal administration and better protection against peripheral tumour via intramuscular administration. In future studies, we would like to investigate whether the synergy of vaccinations with NY-ESO-1 S-FLU virus (intramuscular) and PD-1 antibody-expressing S-FLU (intratumoural injection) could clear the tumour completely (*Hamilton et al., 2018*; *Newman et al., 2020*).

## Materials and methods
### Mice

Human leukocyte antigen (HLA)-A2.1 transgenic mice (*Choi et al., 2003*) (HHD mice) and 1G4 transgenic mice expressing the HLA-A2/NY-ESO-1$_{157-165}$-specific TCR (*Shenderov et al., 2021*) were bred in the local animal facility under specific pathogen-free conditions and used at 6–10 weeks of age. Then, 6- to 7-week-old BALB/c mice were purchased from Envigo. Animal studies have been conducted in accordance with the approval of the United Kingdom Home Office. All procedures were done under the authority of the appropriate personal and project licences issued by the United Kingdom Home Office license number PBA43A2E4.

### Cell line and recombinant virus generation

HEK, 4T1, and MDCK-SIAT1 cells used in this study were obtained from ATCC. The cell lines were regularly checked for the presence of mycoplasma and confirmed mycoplasma free. MDCK-SIAT1, 4T1 and 293T cells were authenticated by STR sequencing. HEK and MDCK-SIAT1 cells were maintained in DMEM, 4T1 cells were maintained in RPMI (Gibco). Media were supplemented with 10% fetal bovine serum (FBS), 2 mM glutamine and penicillin/streptomycin. NY-ESO-1 S-FLU was generated as previously described (*Powell et al., 2012*) with minor modifications. In brief, the codon-optimized cDNA encoding NY-ESO-1 flanked by *NotI* and *EcoRI* cloning sites was synthesized by GeneArt and ligated into the modified S-FLU expression plasmid pPol/S-UL between the 3' and 5' HA packaging sequences. The inactivated HA signal sequence (part of the packaging signal) was optimized by removal of unwanted ATG sequences. GFP S-FLU was made similarly with GFP replacing the NY-ESO-1 sequence. Recombinant NY-ESO-1 S-FLU on the A/PR/8/1934 background were produced by transfection of HEK 293T cells as described (*Powell et al., 2012*; *Fodor et al., 1999*) and cloned twice by limiting dilution in MDCK-SIAT1 cells stably transfected to express coating haemagglutinin from A/PR/8/34 (GenBank accession no. CAA24272.1) to provide the pseudotyping haemagglutinin in trans in viral growth media (VGM; DMEM with 1% bovine serum albumin [Sigma-Aldrich A0336], 10 mM HEPES buffer, penicillin [100 U/ml], and streptomycin [100 µg/ml]) containing 0.75 µg to 1 µg/ml of TPCK-Trypsin (Thermo Scientific, 20233). The NY-ESO-1 S-FLU (PR8) was harvested after 48 hr by centrifugation (1400 × *g* for 5 min) of the culture supernatant to remove debris and kept as a seed virus. NY-ESO-1 S-FLU (X31) was generated by infecting MDCK-SIAT1 stably transfected with X31 H3 (MDCK-X31) with the NY-ESO-1 S-FLU (PR8) seed virus (approximately multiplicity of infection [MOI] 0.01) and the supernatant was harvested as described above after 48 hr. NY-ESO-1 S-FLU was titrated as TCID50 as previously described (*Powell et al., 2012*; *Powell et al., 2019*). In brief, supernatant containing NY-ESO-1 S-FLU was titrated in 1/2-log serial dilution in VGM (total 50 µl) across a flat-bottom 96-well plate seeded with 3e4 MDCK-PR8 or MDCK-X31 cells. After 1 hr incubation at 37°C, 150 µl of VGM containing 1 µg/ml TPCK-Trypsin was added

to each well, and the plate was incubated at 37°C for 48 hr. Then, the plates were washed twice with PBS and fixed with 100 μl of 10% formalin in PBS for 30 min at 4°C and permeabilized with 50 μl of permeabilization buffer (PBS, 20 mM glycine, 0.5% Triton-X100) for 20 min at room temperature. The plates were then washed twice with PBS and stained with 50 μl of PBS containing 0.1% BSA (PBS/0.1% BSA) and mouse anti-NY-ESO1 antibody (Cat# 1:250) for 1 hr. The plates were then washed twice with PBS and stained with 50 μl goat-anti-human AF647 for 1 hr. The plates were then washed with PBS twice and fluorescence was measured using a ClarioStar Plate Reader (BMG Labtech).TCID50 was calculated using the Reed and Muench method (*Reed and Muench, 1938*).

The full-length NY-ESO-1 gene (GenBank no: NM001327) was cloned into the replication-deficient human adenovirus serotype 5 (AdHu5) vector backbone under the control of the human CMV immediate early long promoter to generate the HuAd5-NY-ESO-1 construct. The replication-deficient adenoviral vectors were scaled up by the Viral Vector Core Facility at the Jenner Institute (Oxford, UK) in 293A cells with purification by caesium chloride centrifugation and stocks were stored at –80°C in PBS. Virus titre was determined in a cytopathic effect assay (*Bolinger et al., 2013*). Purity and sterility were confirmed by PCR and inoculation in TSB broth, respectively.

## Virus infections

Influenza virus infections via intranasal route (50 μl volume) were performed with $1 \times 10^6$ TCID$_{50}$ of GFP S-FLU, $1 \times 10^6$ or $3 \times 10^6$ TCID$_{50}$ of NY-ESO-1 S-FLU (PR8) virus or NY-ESO-1 S-FLU (X31) virus and $3.2 \times 10^4$ TCID$_{50}$ of X31 virus. For intramuscular route, $6 \times 10^6$ TCID$_{50}$ (100 μl volume) of NY-ESO-1 S-FLU (PR8) virus was used under inhalation isoflurane anaesthesia. In some experiments, intramuscular infection with $1 \times 10^9$ PFU of Hu Ad5 NY-ESO-1 virus or $5 \times 10^7$ PFU of Fowl Pox NY-ESO-1 virus was also performed.

## Lung cell preparation and CD8+ T cell enrichment

Single-cell preparation from lungs was prepared as described before (*Kandasamy et al., 2016*). Briefly, mice lungs were perfused with 10 ml of PBS and digested in 0.5 mg/ml of collagenase type IV in HBSS/10% FBS for 45 min after chopping finely with scissors. After digestion, lung tissues were passed through a 19G needle a few times and filtered through a 70 μm cell strainer. After two washes in FACS buffer (PBS containing 1% FBS and 2 mM EDTA), the cells were subjected to RBC lysis (QIAGEN, RBC lysis buffer) for 5–7 min followed by two washes with FACS buffer. For adoptive transfer experiments, naïve CD8+ T cells from 1G4 mice spleen were enriched using MACS beads (Pan T cell isolation kit II and CD8a isolation kit, Miltenyi Biotec) and labelled with 5 μM CellTrace Violet (CTV) (Thermo Fisher Scientific) following the manufacturer's instructions. Approximately $2 \times 10^6$ CD8+ T cells in 200 μl volume were adoptively transferred to HHD mice.

## Ex vivo peptide restimulation assay, intracellular staining, and tetramer staining

Splenocytes or lung cells ($2 \times 10^6$) were isolated from either naïve or infected or tumour-bearing HHD or BALB/c mice and were cultured in the presence of HLA-A2-restricted IAV M1 peptide $_{158-66}$(GILGFVFTL) (Cambridge peptide) or NY-ESO-1 peptide $_{157-65}$(SLLMWITQC) (Sigma peptide) or H-2K$^d$ binding IAV NP peptide $_{147-55}$(TYQRTRALV) (Cambridge peptide) or H-2D$^d$-restricted NY-ESO-1 peptide $_{81-88}$(RGPESRLL) (Cambridge peptide) in complete RPMI 1640 medium supplemented with 10% FBS, 2.1 mmol/l ultra-glutamine in the presence of brefeldin A (5 μg/ml) and monensin (2 μM; BioLegend). After 5 hr of incubation, cells were stained for extracellular markers (CD3e, CD8a, B220, and CD44) and viability dye Near IR dead cell staining kit, Invitrogen. Cells were then stained for intracellular IFNγ, TNFα, and IL2 using an Intracellular Fixation and Permeabilization Buffer Set (eBioscience) following the manufacturer's instructions. In some experiments, lung cells or splenocytes were prepared as described above and stained with appropriate surface antigens followed by PE-labelled H-2D$^d$-restricted NY-ESO-1 peptide $_{81-88}$(RGPESRLL) and brilliant violet 421-labelled HLA-A2-restricted IAV M1 peptide $_{158-66}$(GILGFVFTL) or H-2K$^d$-binding IAV NP peptide $_{147-55}$(TYQRTRALV) (kindly provided by NIH tetramer facility at Emory University). Samples were acquired on a FACScanto II or LSR Fortessa-X50 flow cytometer (BD Biosciences), and data were analysed with FlowJo version 10.4.1.

Compensation beads (eBioscience) were used to generate the compensation matrix, and FMOs were used as control.

## Tumour model

For tumour model, 4T1 MBC and 1F4 (tumour cells generated from methylcholanthrene-induced tumours from HHD mice) – syngeneic mouse models have been used as described before (*Shenderov et al., 2021*; *Liu et al., 2019*; *Kim et al., 2009*) with some modifications. For prophylactic lung metastasis model, BALB/c mice were intranasally infected with $1 \times 10^6$ $TCID_{50}$ of GFP S-FLU or NY-ESO-1 S-FLU on day 0 followed by booster infection on day 14. Mice were challenged with tail vein injection of $2 \times 10^5$ 4T1-NY-ESO-1 cells on day 16, and on day 30 (day 14 post injection of tumour cells) mice were euthanized and lungs were fixed in Bouin's solution to count the tumour nodules. For therapeutic tumour model, mice were first injected with $2 \times 10^5$ 4T1-NY-ESO-1 cells on day 0 followed by intranasal infection with GFP S-FLU or NY-ESO-1 S-FLU virus on days 2 and 10 in the doses mentioned above. Mice were closely monitored and euthanized on day 16 post tumour cell injection. For the spontaneous metastasis model, 4T1 cells were subcutaneously injected into the flank of BALB/c mice and the mice were intranasally or intramuscularly infected with GFP S-FLU or NY-ESO-1 S-FLU virus on days 4 and 18 post tumour cell injection. In some experiments, immunized mice also received intraperitoneal injection of anti-PD-1 antibody (12.5 mg/kg, RMP1-14, BioXCell) or isotype antibody. Tumour volumes were measured every other day until days 28–30 at humane end point (when tumour size reaches >1.2 cm³). Then mice were euthanized and metastasis colony formation assay (clonogenic assay) was performed to quantify 4T1-derived cells in the lungs of transplanted mice. Entire lungs were minced and incubated in HBSS media with 10% FBS, supplemented with collagenase (1 mg/ml; Sigma-Aldrich) and deoxyribonuclease (100 µg/ml; Sigma-Aldrich) for 45 min. Thereafter, lung fragments were homogenized through a 100 µm filter and rinsed with 5 ml of PBS. After centrifugation, cell pellets were subjected to RBC lysis (QIAGEN, RBC lysis buffer), and after two times washing in PBS, cells were resuspended in 10 ml of selection media (RPMI 1640 media with 10% FBS and 60 µM 6-thioguanine (Sigma-Aldrich)) before being diluted 1:10 and 1:100 in selection medium. Cells were plated in culture dishes, and the colonies were stained after 14 days with 1% crystal violet/70% methanol solution and counted.

To analyse whether memory T cell response elicited by NY-ESO-1 S-FLU infection protects mice against tumour challenge, HHD mice were infected on day 0 with GFP-S FLU virus or NY-ESO-1 S-FLU virus via intramuscular route and/or NY-ESO-1 S-FLU virus via intranasal route. Mice received booster infection with appropriate virus on day 14 post primary infection. On day 30 post booster infection, mice were challenged with $5 \times 10^5$ NY-ESO-1-expressing tumour cells (1F4) by subcutaneous injection and tumour size was monitored until humane end point.

## Statistical analysis

Unpaired, two-tailed Student's *t*-test and one-way ANOVA with Tukey's multiple comparisons were used to calculate statistical significance (Prism, GraphPad).

## Acknowledgements

This work was supported by CRUK (UK) and MRC. S-FLU work was supported by The Townsend-Jeantet Charitable Trust (Charity No 1011770). The funders had no role in study design, data collection and analysis, decision to publish, or preparation of the manuscript. This article is dedicated to our wonderful mentor, Prof. Vincenzo (Enzo) Cerundolo. MK, UG, AT, and VC designed the study and MK, PR, TT, and LL performed experiments. MK analysed the data and wrote the manuscript. MK, UG, PR, TT, LL, GP, and AT edited the manuscript. All authors discussed the results, commented on the manuscript, and agreed on publication

## Additional information

### Funding

| Funder | Grant reference number | Author |
|---|---|---|
| Cancer Research UK | | Matheswaran Kandasamy |
| MRC-Human Immunology Unit | | Vincenzo Cerundolo |

The funders had no role in study design, data collection and interpretation, or the decision to submit the work for publication.

### Author contributions

Matheswaran Kandasamy, Conceptualization, Investigation, Writing – original draft, Writing – review and editing; Uzi Gileadi, Conceptualization, Supervision, Project administration, Writing – review and editing; Pramila Rijal, Tiong Kit Tan, Gennaro Prota, Investigation, Writing – review and editing; Lian N Lee, Resources, Investigation, Writing – review and editing; Jili Chen, Investigation; Paul Klenerman, Resources, Supervision; Alain Townsend, Conceptualization, Resources, Investigation, Writing – review and editing; Vincenzo Cerundolo, Conceptualization, Resources, Supervision, Investigation, Project administration

### Author ORCIDs

Matheswaran Kandasamy http://orcid.org/0000-0001-7734-4600
Uzi Gileadi http://orcid.org/0000-0001-7348-9204

### Ethics

Animal studies have been conducted in accordance with the approval of, the United Kingdom Home Office. All procedures were done under the authority of the appropriate personal and project licenses issued by the United Kingdom, Home Office License number PBA43A2E4.

### Decision letter and Author response

Decision letter https://doi.org/10.7554/eLife.76414.sa1
Author response https://doi.org/10.7554/eLife.76414.sa2

## Additional files

### Supplementary files

• Transparent reporting form

### Data availability

All data generated or analysed during this study are included in the manuscript and supporting files; Source Data files have been provided for Figures 1 - 7.

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
