## [Editor Report]

The authors found out that virus-based tumour vaccines can induce a robust CTL response capable of limiting tumour progression, which is interesting to many researchers who are looking for ways to enhance CTL response to tumour immunity in combination with checkpoint inhibitors.

---

## [Decision Letter]

**Decision letter after peer review:**

Thank you for submitting your article "Recombinant single-cycle influenza virus with exchangeable pseudo types, allows repeated immunisation to augment antitumour immunity with immune checkpoint inhibitors" for consideration by *eLife*. Your article has been reviewed by 2 peer reviewers, and the evaluation has been overseen by a Reviewing Editor and Päivi Ojala as the Senior Editor. The reviewers have opted to remain anonymous.

Essential revisions:

1) In Figure 6, what are the mechanisms of superior efficacy of S-Flu-based vaccine compared with Ad5 or Pox-based vaccine? Are quantity and/or quality of NY-ESO1-specific CTL different depending on the kinds of virus vectors? Alternatively, other types of immune cells may be also stimulated by each virus vector.

2) The authors should perform additional experiments in which they examine tumor volume following challenge of mice that were influenza infected 30 or more days previously. This experiment would allow the authors to determine the extent to which the reduction in tumor volume seen in their current experiments following challenge infection is due to an ongoing effector response. This is an important proof of concept that a recombinant influenza infection approach generates a memory T cell response capable of eliciting protective immunity against tumor challenge.

3) The author should also assess whether PD1 blockade is able to reduce tumor development when administered to mice that were influenza infected 30 or more days previously and then challenged. This would determine whether PD1 blockade enhances the functionality of memory T cells following recombinant influenza infection. Currently, the authors only perform PD1 blockade beginning 4 days after influenza infection.

Both reviewers recognize that this manuscript is potentially interesting to tumor-immunity researchers. But, reviewers request the experiments (the above-mentioned are essential experiments) to make the impact strong enough for publication in *eLife*.

*Reviewer #1 (Recommendations for the authors):*

1. It is extremely important to enhance the efficacy of ICI with vaccination, while some clinical trials using peptide-based CT antigen vaccine failed to show its synergistic effect. Thus, Flu-based vaccines seem to be a promising strategy to enhance the effect of ICI. However, it is unclear whether the synergistic effect of S-Flu-NYESO1 is not usually observed in pre-clinical studies using other types of vaccine. The authors are recommended to discuss these issues.

2. Is the expression level of NY-ESO1 in 4T1-NY-ESO1 cells comparable to those expressed in primary tumor cells? This may be a difficult question to answer, but the authors are recommended to discuss whether NY-ESO1 is expressed at physiological levels.

---

## [Author Response]

Essential revisions:1) In Figure 6, what are the mechanisms of superior efficacy of S-Flu-based vaccine compared with Ad5 or Pox-based vaccine? Are quantity and/or quality of NY-ESO1-specific CTL different depending on the kinds of virus vectors? Alternatively, other types of immune cells may be also stimulated by each virus vector.

It is an interesting question and thank you for your observation. A significantly higher CTL response with S-FLU virus compared to POX virus explains the enhanced anti-tumour effect of S-FLU over Pox virus based vaccine. However, we don’t know the clear reason for the superior anti-tumour efficacy of S-FLU -NY-ESO1 over Ad5 based vaccine. The frequencies of NY-ESO-1 specific CTL responses in lungs and spleen were similar following S-FLU-NY-ESO1 or Hu-Ad5 NY-ESO1 administration. We haven’t investigated the quality or polyfunctionality of CTLs and differential cellular infiltration to the tumour site post immunization with S-FLU or Ad5virus. S-FLU and Ad5 virus elicit different kinds of activation of innate signalling ( S-FLU is sensed by TLR3 and 7 and Ad5 virus is sensed by TLR2,4 and 9) and we surmise that these different innate immune signalling may influence subsequent immune response including cellular infiltration.

2) The authors should perform additional experiments in which they examine tumor volume following challenge of mice that were influenza infected 30 or more days previously. This experiment would allow the authors to determine the extent to which the reduction in tumor volume seen in their current experiments following challenge infection is due to an ongoing effector response. This is an important proof of concept that a recombinant influenza infection approach generates a memory T cell response capable of eliciting protective immunity against tumor challenge.

Thank you for your suggestion. We investigated whether the memory T cell response can elicit protective immunity against tumour challenge in HHD mice. We have included the result as supplementary Figure 10 and updated the text in manuscript.

3) The author should also assess whether PD1 blockade is able to reduce tumor development when administered to mice that were influenza infected 30 or more days previously and then challenged. This would determine whether PD1 blockade enhances the functionality of memory T cells following recombinant influenza infection. Currently, the authors only perform PD1 blockade beginning 4 days after influenza infection.Both reviewers recognize that this manuscript is potentially interesting to tumor-immunity researchers. But, reviewers request the experiments (the above-mentioned are essential experiments) to make the impact strong enough for publication in eLife.

Thank you for your observation and excellent suggestion. Unfortunately, our lab has been closed because of our PI’s ( Vincenzo Cerundolo ) untimely death. Because of that and on-going COVID-19 pandemic related limitations, we managed to do the experiment for memory T cell response only without PD-1 co-administration.

Reviewer #1 (Recommendations for the authors):1. It is extremely important to enhance the efficacy of ICI with vaccination, while some clinical trials using peptide-based CT antigen vaccine failed to show its synergistic effect. Thus, Flu-based vaccines seem to be a promising strategy to enhance the effect of ICI. However, it is unclear whether the synergistic effect of S-Flu-NYESO1 is not usually observed in pre-clinical studies using other types of vaccine. The authors are recommended to discuss these issues.

Enhanced anti-tumour effect of virus-based vaccine co-administered with ICI has been recently reported for adenovirus based vaccine ( McAuliffe J et al., J Immunother Cancer. 2021 Sep;9(9): e003218) and S-FLU-NY-ESO-1 virus has not been investigated before for anti-tumour therapy and in this study, we report the synergistic effect of S-FLU-NY-ESO-1 with immune check point inhibitor for the first time. We included these issues in discussion.

2. Is the expression level of NY-ESO1 in 4T1-NY-ESO1 cells comparable to those expressed in primary tumor cells? This may be a difficult question to answer, but the authors are recommended to discuss whether NY-ESO1 is expressed at physiological levels.

The level of NY-ESO-1 expression may vary in different types and different stages of tumour. The physiological level of NY-ESO-1 expression in melanoma is an average of ∼10-50 NY-ESO-1 _157-165_ Ags per cell in which density most self Ags and TAAs are presented (Purbhoo MA et al., J Immunol. 2006 Jun 15;176(12):7308-16). We used 4T1 tumour cells with lower level of NY-ESO-1 expression for our tumour study since the higher expression of NY-ESO-1 in tumour cells resulted in spontaneous tumour rejection. We included this information in discussion.